# p53 in Proximal Tubules Mediates Chronic Kidney Problems after Cisplatin Treatment

**DOI:** 10.3390/cells11040712

**Published:** 2022-02-17

**Authors:** Shuangshuang Fu, Xiaoru Hu, Zhengwei Ma, Qingqing Wei, Xiaohong Xiang, Siyao Li, Lu Wen, Yumei Liang, Zheng Dong

**Affiliations:** 1Laboratory of Kidney Disease, Department of Nephrology, Hunan Provincial People’s Hospital (The First-Affiliated Hospital of Hunan Normal University), Changsha Clinical Research Center for Kidney Disease, Hunan Clinical Research Center for Chronic Kidney Disease, Changsha 410005, China; fushsh0418@hunnu.edu.cn (S.F.); liangyumei@163.com (Y.L.); 2Department of Cellular Biology and Anatomy, Medical College of Georgia, Augusta University, Augusta, GA 30912, USA; xhu1@augusta.edu (X.H.); zma@augusta.edu (Z.M.); qwei@augusta.edu (Q.W.); xixiang@augusta.edu (X.X.); sli2@augusta.edu (S.L.); luwen@augusta.edu (L.W.); 3Department of Nephrology, The Second Xiangya Hospital, Central South University, Changsha 410001, China; 4Charlie Norwood Veterans Affairs Medical Center, Augusta, GA 30904, USA

**Keywords:** cisplatin, nephrotoxicity, p53, chronic kidney disease, apoptosis, fibrosis

## Abstract

Nephrotoxicity is a major side-effect of cisplatin in chemotherapy, which can occur acutely or progress into chronic kidney disease (CKD). The protein p53 plays an important role in acute kidney injury induced by cisplatin, but its involvement in CKD following cisplatin exposure is unclear. Here, we address this question by using experimental models of repeated low-dose cisplatin (RLDC) treatment. In mouse proximal tubular BUMPT cells, RLDC treatment induced p53 activation, apoptosis, and fibrotic changes, which were suppressed by pifithrin-α, a pharmacologic inhibitor of p53. In vivo, chronic kidney problems following RLDC treatment were ameliorated in proximal tubule-specific p53-knockout mice (PT-p53-KO mice). Compared with wild-type littermates, PT-p53-KO mice showed less renal damage (KIM-1 positive area: 0.97% vs. 2.5%), less tubular degeneration (LTL positive area: 15.97% vs. 10.54%), and increased proliferation (Ki67 positive area: 2.42% vs. 0.45%), resulting in better renal function after RLDC treatment. Together, these results indicate that p53 in proximal tubular cells contributes significantly to the development of chronic kidney problems following cisplatin chemotherapy.

## 1. Introduction

Cisplatin is one of the most effective antitumor chemotherapeutic agents [1]. However, its cytotoxicity to normal cells, especially the nephrotoxicity, limits the use of cisplatin [2,3,4]. Clinically, nearly 30% of patients develop acute kidney injury (AKI) as well as chronic kidney disease (CKD) after cisplatin administration [5]. The proximal tubule is the major renal compartment in the kidneys that is vulnerable to cisplatin nephrotoxicity. In the past decades, cisplatin-induced AKI has been widely examined in cellular and animal models to identify the critical molecular pathways leading to nephrotoxicity, including p53 [6], microRNAs [7,8], oxidative stress [9], and mitochondrial damage [10]. However, the investigation of cisplatin-induced CKD and related mechanisms is limited due to the lack of appropriate research models. Fortunately, cisplatin-induced CKD models have been established recently with repeated low-dose cisplatin (RLDC) treatment that mimics the clinical use of cisplatin [11,12,13,14,15,16,17].

The protein p53 is a well-known tumor suppressor that responds to cellular stress by inducing apoptosis and cell-cycle arrest by transactivating genes, such as PUMA-α and p21 [18,19]. In cisplatin-induced AKI, p53 is induced to promote kidney tubular injury [20,21,22]. Of note, pifithrin-α, a pharmacological inhibitor of p53, can ameliorate cisplatin-induced tubular cell death and AKI [23,24]. Although its role is controversial, p53 is also closely related to renal fibrosis development. Yang et al. [25] revealed that pifithrin-α could alleviate the progression of fibrosis following renal ischemia/reperfusion injury. Zhang et al. [26,27] showed that the suppression of p53 ameliorated renal fibrosis in UUO and diabetic nephropathy models. Furthermore, Ying et al. [28] reported that mice with proximal tubule specific deletion of p53 showed less interstitial fibrogenesis after renal ischemia/reperfusion injury. However, Dagher et al. [29] indicated that the inhibition of p53 with pifithrin-α increased renal fibrosis after renal ischemia/reperfusion injury in rats. Regardless, the role of p53 in cisplatin-induced CKD and fibrosis is unknown.

In the current study, we demonstrate the activation of p53 in RLDC-induced apoptosis and fibrosis in renal tubular cells, and elucidate the effect of pifithrin-α to prove the injurious role of p53 in this model. Further, we report on how we examined the role of p53 in RLDC-induced CKD in vivo by using the proximal tubule-specific p53 knockout mouse model.

## 2. Materials and Methods

### 2.1. Cells and Treatment

The BUMPT (mouse proximal tubular) cell line was originally obtained from Dr. William Lieberthal and Dr. John Shwartz at Boston University [30]. BUMPT cells were seeded at a density of 0.2 million cells/dish in 35 mm dishes to reach 30–40% confluence by the next day for the experiment. Cells were cultured in DMEM medium with 10% fetal bovine serum at 37 °C in 5% CO_2_. Freshly prepared cisplatin and pifithrin-α were added to BUMPT cells at a final concentration of 2 μM and 20 μM in culture medium. For RLDC treatment, BUMPT cells were incubated with 2 μM cisplatin for 7 h daily for 4 days. 20 μM pifithrin-α was added daily for the whole treatment period. At the end of incubation, the cellular and nuclear morphology was analyzed by phase contrast microscopy (Thermo Fisher Scientific, Waltham, MA, USA) and fluorescent microscopy (Carl Zeiss Microscopy, Thornwood, NY, USA). Whole-cell lysate was collected for biochemical analyses, as described in a previous study [23].

### 2.2. Animals and Treatment

PT-p53 KO mouse line was generated by mating p53 (flox/flox) mice from Jackson Laboratory with the PEPCK-Cre mice from Dr. Volker Haase (Vanderbilt University School of Medicine, Nashville, TN, USA), as described previously [6]. Male p53 (flox/flox) mice of 12 weeks with or without PEPCK-Cre gene were used for experiments. The mice were housed under a 12:12-hour light/dark cycle with free access to food and water in the animal facility of the Charlie Norwood VA Medical Center at Augusta, GA, USA. All the animal experiments were conducted in accordance with a protocol approved by the Institutional Animal Committee Care and Use Committees of the Charlie Norwood VA Medical Center at Augusta. The genomic DNA samples were extracted from tail or kidney for PCR-based genotyping. Mice received 8 mg/kg cisplatin or saline (vehicle solution) via intraperitoneal injection once a week for 4 weeks and were euthanized on the 36th day after the initial dosage of cisplatin. Blood and kidney samples were collected for the further analysis. The primer pair for PCR detection of the CRE gene was 5′-ACC TGA AGA TGT TCG CGA TTA TCT-3′ (forward) and 5′-ACC GTC AGT ACG TGA GAT ATC TT-3′ (reverse), while that for p53-floxed allele was 5′-CAC AAA AAC AGG TTA AAC CCA-3′ (forward) and 5′-GAA GAC AGA AAA GGG GAG GG-3′ (reverse).

### 2.3. Reagents and Antibodies

Cisplatin was purchased from Sigma (St. Louis, MO, USA). Pifithrin-α was purchased from Selleck (Houston, TX, USA). Primary antibodies used in this study were from the following sources: rabbit polyclonal anti-cleaved caspase-3, mouse polyclonal anti-p53, and rabbit polyclonal anti-phospho-p53 (Ser15), GAPDH. The β-actin antibodies were from Cell Signaling Technology (Danvers, MA, USA). Rabbit polyclonal anti-fibronectin was from Abcam (Cambridge, MA, USA). Rabbit polyclonal anti-Collagen I and rabbit polyclonal anti-CTGF were from Novus Biologicals (Littleton, CO, USA). Mouse polyclonal anti-p21 was from Invitrogen (Carlsbad, CA, USA). Secondary antibodies were from Jackson ImmunoResearch (West Grove, PA, USA).

### 2.4. Immunofluorescence

Immunofluorescence staining was conducted as described in our previous work [23,24,31]. Briefly, BUMPT cells were seeded on glass coverslips. After treatment, cells were fixed with 4% fresh prepared paraformaldehyde for 30 min. Next, the cells were incubated in a blocking buffer containing 0.4% Triton X-100 and 2% normal goat serum for 1 h, followed by incubation with anti-p53 primary antibody for 1 h. Finally, the cells were exposed to Cy3-labeled goat-anti-mouse secondary antibody (Chemicon, Temecula, CA, USA) for 1 h. The signals were examined by fluorescence microscopy (Carl Zeiss Microscopy, Thornwood, NY, USA).

### 2.5. Immunoblot Analysis

The whole-cell lysates or kidney lysates were collected using a buffer containing 2% SDS buffer with 1% protease inhibitor cocktail and 1% benzonase (Sigma, St. Louis, MO, USA) [23,32]. The protein concentration was measured with the Pierce Bicinchoninic Acid (BCA) reagent (ThermoFisher Scientific, Waltham, MA, USA). An equal amount, usually 20 μg, of protein was loaded for SDS-PAGE and transferred onto PVDF membrane. The membranes were then incubated sequentially with a blocking solution containing 5% nonfat milk, the primary antibody incubation in 4 °C overnight, and finally horseradish peroxidase-conjugated secondary antibody. The antigens on the blots were revealed using the enhanced chemiluminescence (ECL) kit from Bio-Rad Laboratories (Hercules, CA, USA) to record signals by MyECL Imager (ThermoFisher Scientific, Waltham, MA, USA) or KwikQuant Imager (Kindle Biosciences, Greenwich, CT, USA). The densitometry of the blots was performed with ImageJ software (https://imagej.nih.gov/ij/download.html, accessed on 19 August 2021). 

### 2.6. Morphological Examination of Apoptosis

As described in our previous publications [23,33], apoptotic cells were identified by their morphological changes, including nuclear fragmentation and condensation, cellular shrinkage, and blebbing or formation apoptotic bodies. The nuclei were stained by 10 μg/mL of Hoechst 33,342 for 2–5 min. Cellular and nuclear morphology were examined by phase contrast and fluorescence microscopy. Four random fields with about 200 cells/field were examined to estimate the apoptosis percentage.

### 2.7. RNA Isolation and Quantitative Real-Time PCR

As described in our recent publications [32,34,35], the mirVana miRNA Isolation Kit (Thermo Fisher Scientific, Waltham, MA, USA) was used for total RNA extraction from BUMPT cells. For quantitative real-time PCR (qPCR) to analyze gene expression, 1 μg total RNA was reversely transcribed using a cDNA Transcription Kit (Bio-Rad Laboratories, Hercules, CA, USA), and qPCR was performed using SYBR Green PCR Master Mix (Bio-Rad Laboratories, Hercules, CA, USA). The mRNA level of CTGF was normalized by GAPDH. The primer pair for CTGF was 5′-GTT ACC AAT GAC AAT ACC TTC TGC-3′ (forward) and 5′-TTG ACA GGC TTG GCG ATT-3′ (reverse). The primer pair for GAPDH was 5′-ACG GCA CAG TCA AGG CTG AG-3′ (forward) and 5′-GGA GGC CAT GTA GAC CAT GAG G-3′ (reverse). Fold change in expression was quantified using ΔCt values.

### 2.8. Renal Function and Transcutaneous Measurement of GFR

BUN was measured using the commercial kit from Stanbio Laboratory to monitor renal function, as previously described [17,36]. Serum creatinine was measured based on Jaffe reaction, as previously described [34]. GFR was assessed by transcutaneous measurement of FITC-labeled sinistrin excretion [16]. In brief, mice were anesthetized with isoflurane and part of the back skin was depilated. The transdermal GFR monitor (MediBeacon, Mannheim, Germany) was fixed on the depilated skin using a double-sided adhesive patch to measure the background signal for 2 min. Nexy, FITC-sinistrin (15 mg/mL dissolved in 0.9% sterile saline) was injected in the retro-orbital sinus, and GFR was monitored for 60 min. The data were analyzed using elimination kinetics curve of FITC-sinistrin.

### 2.9. Histological Staining

Kidney samples or cells were fixed with 4% paraformaldehyde. Paraffin-embedded renal sections were used for immunohistochemical staining or Sirius Red staining. For Sirius Red staining, the sections were rehydrated and stained with Sirius Red/Fast Green Collagen Staining Kit from Chondrex (Woodinville, WA, USA). For immunohistochemical staining, after rehydration and blocking, the kidney sections were exposed to primary antibody (anti-p53, anti-KIM-1 or anti-Ki67) overnight at 4 °C, followed by incubation with a secondary antibody. For p53 staining, after washing, the slides were blocked again using avidin-biotin blocking reagent (SP-2001, Vector Laboratories), followed by incubation with 1:1000 biotinylated donkey anti-rabbit secondary antibody (AP182B, Millipore, Burlington, MA, USA) for 1 h at room temperature. After signal amplification using Tyramide Signal Amplification Biotin System (Perkin Elmer, NEL700A001KT, Waltham, MA, USA), the sections were incubated with a VECTASTAIN^®^ ABC kit (PK-6100, Vector Laboratories, Burlingame, CA, USA) and color was developed with a DAB kit (SK-4100, Vector Laboratories, Burlingame, CA, USA). For KIM-1 staining, the slides were developed with an Alkaline Phosphatase (AP) Substrate Kit (Vector Laboratories, Burlingame, CA, USA). For Ki67 staining, the slides were developed with a VECTASTAIN Elite ABC Kit and ImmPACT DAB Peroxidase Substrate (Vector Laboratories, Burlingame, CA, USA). For proximal tubules identification, FITC-conjugated LTL (dilution 1:100, Vector Laboratories, Burlingame, CA, USA) were added after secondary antibody incubation and AP development. The stained cells or kidney tissues were mounted with ProLong Gold Antifade Mountant (Invitrogen, Carlsbad, CA, USA) or Permount Mounting Medium (Thermo Fisher Scientific, Waltham, MA, USA), and examined by bright field histology microscopy (Olympus Optical Co., Ltd., Tokyo, Japan) or fluorescent microscopy (Carl Zeiss Microscopy, Thornwood, NY, USA).

### 2.10. Image Quantification

To quantitatively estimate the positive areas of p53, Sirius Red, Kim-1, LTL, and Ki67 staining, 10 fields (100× magnification) were randomly selected from each slide and analyzed by Image J software (https://imagej.nih.gov/ij/download.html, accessed on 18 December 2021). These 10 fields covered most of the area of the kidney specimen examined. After setting the threshold for density selection, the software automatically calculated the positive signal area percentage, and the average of these 10 fields was used for statistical analysis.

### 2.11. Statistics Analysis

All experiments in this study were repeated at least three times. Data were expressed as means ± SD. Statistical analysis was conducted using GraphPad Prism 9.0 software. Statistical differences in multiple groups with one variable (Figure 1, Figure 2 and Figure 3) were assessed by one-way ANOVA, while those with two or more variables (Figure 4, Figure 5 and Figure 6) were assessed by two-way ANOVA. *p <* 0.05 was considered statistically significant.

## 3. Results

### 3.1. Activation of p53 in RLDC-Treated BUMPT Cells and Its Inhibition by Pifithrin-α

To examine the involvement of p53 in repeated low-dose cisplatin (RLDC)-induced kidney injury, we initially tested the effect of pifithrin-α, a pharmacological inhibitor of p53 that blocks its transcriptional activity [37]. In this test, we used the in vitro model of RLDC injury by incubating mouse kidney proximal tubular (BUMPT) cells with 2 μM cisplatin for 7 h every day for 4 days [16]. Both immunofluorescence and immunoblot analysis showed that p53 expression was very low in control cells, and was significantly upregulated by RLDC (Figure 1A–C). The induction of p53 during RLDC treatment was partially suppressed by pifithrin-α. Consistently, RLDC induced p53 phosphorylation, shown by Ser-15 phosphorylated-p53, which was also suppressed by pifithrin-α (Figure 1B,D). We further analyzed the transcriptional activity of p53 by examining p21, a downstream transcriptional target of p53 [38]. The protein p21 was induced during RLDC treatment, and this induction was again suppressed by pifithrin-α (Figure 1B,E). The inhibitory effects of pifithrin-α on p53 the accumulation, phosphorylation, and transcriptional activity was dose-dependent in the range of 5–20 μM. These results indicate that pifithrin-α may inhibit p53 in the RLDC model by suppressing p53 accumulation and blocking its transcriptional activity.

### 3.2. Pifithrin-α Attenuates Apoptosis in RLDC-Treated BUMPT Cells

In acute nephrotoxicity induced by high doses of cisplatin, p53 plays an important role in kidney tubular cell apoptosis [23,24]. Furthermore, p21, which was regulated by p53 in the RLDC model (Figure 1B,E), contributes to the regulation of apoptosis in acute cisplatin nephrotoxicity [6,20,39]. Therefore, we determined whether p53 inhibition by pifithrin-α could suppress apoptosis in RLDC-treated BUMPT cells. As shown in Figure 2, we observed ~1% apoptosis in the control groups. After RLDC treatment, apoptosis increased to ~10%, which was suppressed to 5% by pifithrin-α (Figure 2A,B). Consistently, immunoblot analyses showed that pifithrin-α suppressed caspase activation, as indicated by the appearance of cleaved or active caspase 3 (Figure 2C,D).

### 3.3. Pifithrin-α Reduces RLDC-Induced Fibrotic Changes in BUMPT Cells

In previous work, RLDC-treated BUPMT cells changed their morphology to a spindle shape [16]. This morphological change was verified in the current study, and, moreover, we found that pifithrin-α could ameliorate the morphological change (Figure 2A). We further explored the effect of p53 inhibition on the expression of extracellular matrix (ECM) proteins in this model. The immunoblot analysis showed that two ECM proteins, fibronectin and collagen I, increased significantly after RLDC treatment. Pifithrin-α treatment obviously reduced the fibronectin and collagen I expression (Figure 3A–C). Connective tissue growth factor (CTGF), a pro-fibrotic growth factor, plays an important role in cisplatin induced fibrosis [40,41] and can be regulated by p53 [40]. We examined the effect of pifithrin-α on CTGF. The immunoblot analysis showed that the expression of CTGF increased significantly after RLDC treatment (Figure 3D,E), and that this induction was blocked by pifithrin-α (Figure 3D,E). Consistently, RLDC increased the mRNA of CTGF to nearly two-fold compared with the control, and this induction was prevented by pifithrin-α (Figure 3F). Therefore, the inhibition of p53 by pifithrin-α can ameliorate RLDC-induced fibrotic changes in BUMPT cells, including morphology, ECM protein accumulation, and the expression of pro-fibrotic cytokines.

### 3.4. RLDC-Induced Renal Fibrosis Is Attenuated in PT-p53-KO Mice

To further determine the role of p53 in RLDC-induced kidney injury, we tested the conditional knockout mice with p53 specifically knocked out in kidney proximal tubule cells (PT-p53-KO) [6]. The PT-p53-KO mice and their wild-type littermates (PT-p53-WT) were given 4 weekly injections of 8 mg/kg cisplatin (with saline as the solution control). The mice were euthanized on the 36th day after the first cisplatin injection to collect samples for analysis (Figure 4A). To verify the genotype and PT-p53 depletion, the genomic DNA samples from the kidney cortex were examined by two sets of PCR for each mouse, indicating the existence of the CRE gene and recombinant p53 allele in the PT-p53-KO mice (Figure 4B). We further analyzed p53 expression in kidney tissues by immunohistochemistry (Figure 4C). In control kidneys, very few cells had p53 staining in their nuclei. After RLDC injury, wild-type kidneys showed a marked increase in renal tubular cells with nuclear p53 staining, which was significantly reduced in the PT-p53-KO kidneys. The quantification of p53 positive cells further verified this conclusion (Figure 4C). We then examined the expression of fibrosis-related proteins (fibronectin, collagen I, vimentin and CTGF), apoptosis-related protein (cleaved caspase3), and p21 in those kidneys by immunoblotting. After RLDC, there was an obvious accumulation of fibrosis-related proteins in the kidneys of the wild-type mice, whereas this accumulation was significantly attenuated in the PT-p53-KO mouse kidneys (Figure 4D). The expression of p21 was suppressed in the PT-p53-KO kidneys compared with the PT-p53-WT kidneys (Figure 4D). However, no obvious change in cleaved caspase 3 was detected, possibly because these kidneys had recovered from the acute tubular injury at tissue collection (Figure 4D). Moreover, Sirius Red staining indicated the significant accumulation of collagens in the kidneys after cisplatin treatment, which was obviously reduced by PT-p53 knockout (Figure 4E). Overall, PT-p53-KO mainly attenuated renal fibrosis following RLDC treatment.

### 3.5. RLDC-Induced Kidney Atrophy and Renal Function Decline Are Moderately Suppressed in PT-p53-KO Mice

To further determine the involvement of p53 in chronic kidney problems induced by RLDC, we analyzed the kidney weight/body weight ratio, blood urea nitrogen (BUN), serum creatinine, and glomerular filtration rate (GFR). As shown in Figure 5A,B, RLDC induced kidney atrophy in both PT-p53-WT and KO mice in comparison to the control kidneys. By kidney weight/body weight ratio, PT-53-KO mice generally showed less kidney atrophy but the difference was not statistically significant due to the data variation especially in PT-p53-KO mice (Figure 5B). BUN and serum creatinine levels increased after RLDC treatment without showing significant differences between the PT-p53-WT and KO mice (Figure 5C,D). To measure renal function, we determined the clearance of FITC-sinistrin to indicate GFR. RLDC induced significant decreases in GFR, and this decrease was partially but significantly prevented in the PT-p53-KO mice (Figure 5E,F). Taken together, these results indicate that p53 deficiency in proximal tubules may partially prevent or slow down the development of chronic kidney problems after RLDC treatment.

### 3.6. Proximal Tubular p53 Deficiency Suppresses Tubular Degeneration and Induces Renal Cell Proliferation in RLDC-Treated Mice

Renal tubular degeneration contributes significantly to kidney atrophy [42]. Thus, we examined the expression of kidney injury marker (KIM-1) and proximal tubules marker (lotus tetragonolobus lectin, LTL), respectively, to identify the degenerated proximal tubules. With RLDC administration, the degenerating proximal tubules lost LTL staining due to a loss of brush border, which was accompanied with enhanced KIM-1 signal in these tubules (Figure 6A). This proximal tubular degeneration was obviously repressed in the PT-p53-KO mice (Figure 6A,C,D). Ki67 is a marker of cell proliferation. Following RLDC treatment, some Ki67-positive cells appeared in the PT-p53-WT kidneys, and much more Ki67 positive cells showed up on the kidneys of PT-p53-KO mice, especially at the junctional area between the renal cortex and the medulla (Figure 6B,E). Overall, p53 deficiency suppresses RLDC-associated renal proximal tubular degeneration and induces renal cell proliferation.

## 4. Discussion

Using the pharmacological inhibitor pifithrin-α and PT-p53-KO mouse model, this study provides the first evidence for the involvement of p53 in cisplatin-induced chronic kidney problems or CKD. The protein p53 was activated in RLDC-treated BUMPT cells and induced apoptosis and profibrotic changes in these cells (Figure 1, Figure 2 and Figure 3). In vivo, p53 deficiency in proximal tubules clearly reduced renal fibrosis and the production of pro-fibrotic cytokines, such as CTGF (Figure 4). Furthermore, p53 deficiency in proximal tubules reduced tubular degeneration and improved tubular cell proliferation, resulting in reduced tubular atrophy, kidney weight loss, and renal function decline.

In kidneys, fibrosis development after injury is affected by the severity of the initial injury. In this study, we detected p53 activation and apoptosis during RLDC treatment and the protective effect of pifithrin-α against apoptosis in BUMPT cells. In vivo, RLDC treatment induced p53 activation in tubular cells in wild-type mice. However, renal cell apoptosis was not obvious in mice after RLDC treatment. The lack of these changes is probably related to the experiment’s ending at the point at which we collected samples. We collected cultured tubular samples on the second day after the last cisplatin treatment. However, we sacrificed the mice and collected renal samples at about two weeks after the last cisplatin administration in vivo (Figure 4A), when the kidneys might have recovered from the initial apoptosis induction phase. We did not examine the number of apoptotic tubular cells at the acute phase of cisplatin treatment in this study, because the focus of this study was chronic kidney problems after repeated low-dose cisplatin treatment. Thus, it is possible that lower levels of cell injury or death in PT-p53-KO mice could be partially responsible for the ameliorated renal fibrosis in these animals. Future study should use an inducible p53-KO model to induce p53 ablation from renal tubular cells after RLDC treatment.

In addition to regulating renal tubular cell death, p53 may be involved in the regulation of renal fibrosis through the TGF-β pathway [43]. The activation of p53, especially the phosphorylation of Ser15, has been reported to be indispensable in the TGF-β pathway in renal fibrosis development after ureteral obstruction, promoting the upregulation of fibrotic markers, including connective tissue growth factor (CTGF) and fibronectin [43]. In our study, we also detected obvious p53 activation with Ser15 phosphorylation, and its level was closely associated with CTGF and fibronectin induction in vitro (Figure 3). In vivo, we also detected obvious p53 accumulation, which was accompanied by significant upregulation of CTGF and fibronectin in the kidneys after cisplatin treatment, supporting the involvement of p53 (Figure 4).

CTGF was shown to have potent pro-fibrotic properties in various models [40,44]. p53 can induce both CTGF gene transcription and protein translation through the inhibition of microRNAs in the mir-17-92 cluster [40]. In our study, we detected both mRNA and protein changes of CTGF in BUMPT cells associated with p53 activation (Figure 3D,E). Although we are unsure as to the involvement of microRNAs in CTGF synthesis, the overall induction fold of CTGF protein level is much higher than the mRNA fold change after cisplatin treatment (Figure 3E,F), suggesting the existence of post-transcriptional regulation.

The maladaptive repair after injury due to renal tubular cells that enter the cell cycle but are arrested in the G2/M phase has been shown to be a pro-fibrotic mechanism in several models [25,45]. p53 is well known for its function in cell cycle regulation [46,47,48], and the suppression of p53 is anti-fibrotic through the relief of cell cycle arrest [25]. In the current study, we found that p53 depletion from proximal tubule cells obviously enhanced Ki-67 expression after cisplatin injury, indicating an increase in proliferation for kidney repair. Although we did not analyze the cell cycle status of the Ki67-positive cells, the increased proliferation in the PT-p53-KO mice was expected to improve tubular repair, contributing to the observed reduction in kidney atrophy or weight loss in these animals.

In this study, we mainly examined the role of p53 in renal proximal tubular cells in cisplatin-induced CKD. However, more detailed studies are necessary to examine the role of p53 in different types of renal cells to clarify whether p53 can be used as a therapeutic target for cisplatin-induced CKD treatment. First, pifithrin-α has distinct or opposite roles in the pathogenesis of fibrosis in different types of cells and models [25,26,29]. Our initial test on the use of pifithrin-α to globally inhibit p53 did not consistently relieve renal fibrosis in vivo. In addition, we noticed that the proximal tubular p53-deficient mice used in our experiments showed diverse sensitivities to cisplatin injury. Compared with the wild-type mice, some PT-p53-KO mice were extremely sensitive to cisplatin injury and showed medium-to severe-renal injury after RLDC treatment. In our previous study, examining p53 in cisplatin-induced AKI [24], we also noticed that two groups of global p53 knockout mice had different sensitivities to cisplatin injury, while wild-type mice showed relatively consistent levels of injury. The reason for the injury sensitivity differences in the p53 mice is still unclear. One possibility is that p53 is a tumor suppressor and its depletion may cause some spontaneous malignancy in kidneys, which will affect the severity of cisplatin-induced injury.

It is well-known that p53 plays a critical role in tumor suppression, mainly by inducing growth arrest, senescence, and apoptosis in cancer cells, as well as by blocking angiogenesis in tumors [49]. p53 is also associated with the anti-tumor effect of cytotoxic anticancer drugs, such as cisplatin. Therefore, the anti-tumor effect of cisplatin is expected to be reduced when p53 is inhibited or absent. In this regard, blocking p53 may not be an ideal way approach to kidney protection during cisplatin chemotherapy, because this may reduce the anti-cancer effect of cisplatin. Nonetheless, p53 is mutated or deficient in about half of cancers [50], while p53 inhibitors may protect the kidneys without reducing the anti-cancer effect of cisplatin.

In conclusion, in this study, we confirm the pivotal role of p53 in renal proximal tubular cells in cisplatin-induced CKD. The underlying mechanisms may involve the induction of renal cell apoptosis, the enhanced expression of fibrotic factors, and suppressed renal repair after p53 activation (Figure 7). Further exploration of p53’s function in other renal cells may help to support the therapeutic potential of p53 inhibition in cisplatin-induced CKD.

## Figures and Tables

**Figure 1 cells-11-00712-f001:**
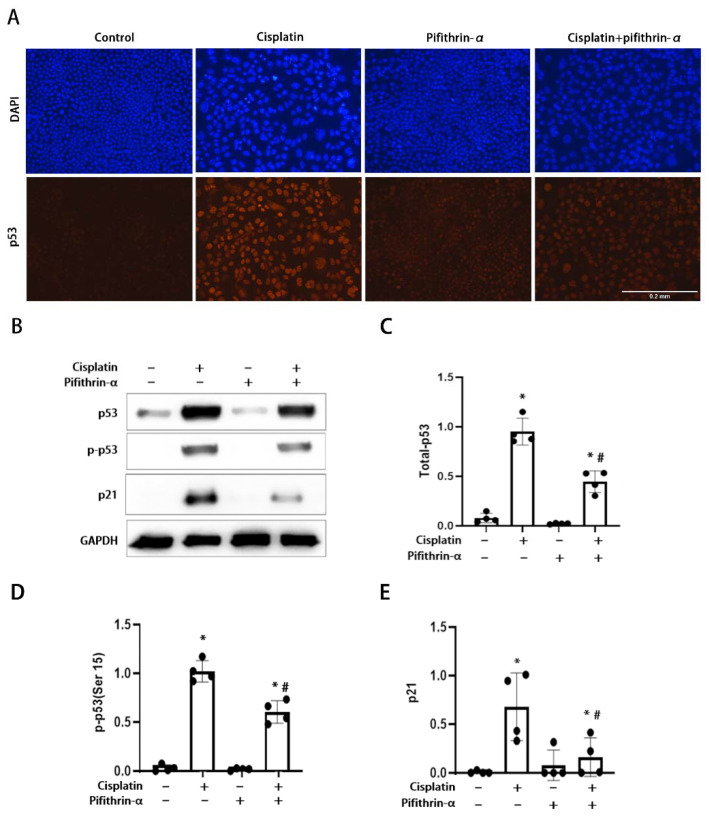
Activation of p53 in RLDC-treated BUMPT cells and its inhibition by pifithrin-α. BUMPT cells were incubated with 2 μM cisplatin for 7 h daily for 4 days in the absence or presence of 20 μM pifithrin-α. (**A**) p53 immunofluorescence and DAPI staining of nuclei. Scale bar = 0.2 mm. (**B**) Immunoblots of p53, phospho-p53 (Ser 15), and p21 with GAPDH as internal loading marker. (**C**–**E**) Densitometric analysis of p53, p-p53 (Ser15) and p21 normalized to GAPDH. Data are expressed as means ± SD (n = 4). * *p* < 0.05, significantly different from the control group. ^#^
*p* < 0.05, significantly different from the cisplatin only group.

**Figure 2 cells-11-00712-f002:**
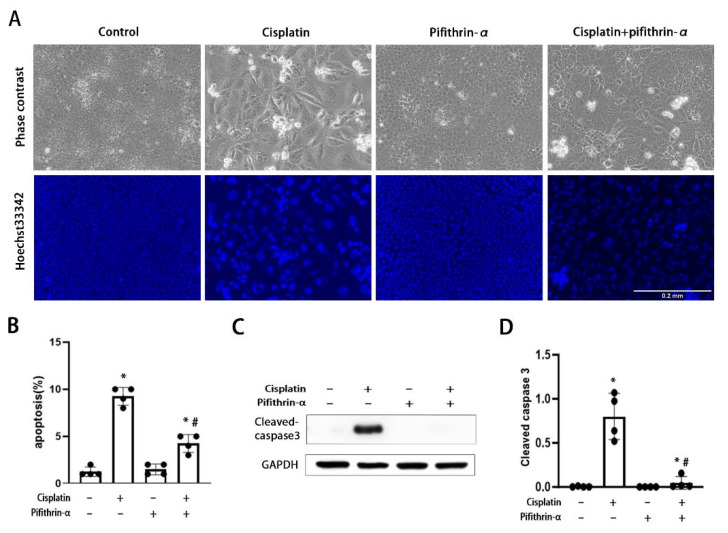
Pifithrin-α attenuates apoptosis during RLDC-treatment of BUMPT cells. BUMPT cells were incubated with 2 μM cisplatin for 7 h daily for 4 days in the absence or presence of 20 μM pifithrin-α. (**A**) Cell morphology and Hoechst 33,342 staining of nuclei. Scale bar = 0.2 mm. (**B**) Percentage of apoptosis assessed by morphology (n = 4). (**C**) Immunoblots of cleaved-caspase 3 and the internal control GAPDH. (**D**) Densitometric analysis of cleaved caspase3 normalized to GAPDH. Data are expressed as mean ± SD (n = 4). * *p* < 0.05, significantly different from the control group. ^#^
*p* < 0.05, significantly different from the cisplatin-only group.

**Figure 3 cells-11-00712-f003:**
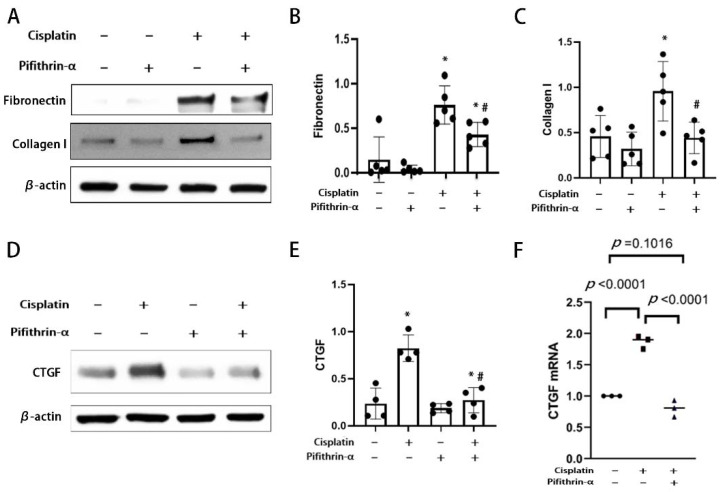
Pifithrin-α reduces RLDC-induced fibrotic changes in BUMPT cells. BUMPT cells were incubated with 2 μM cisplatin for 7 h daily for 4 days in the absence or presence of 20 μM pifithrin-α. Whole cell lysate was collected for immunoblot analysis of fibronectin, collagen I, CTGF and β-actin. (**A**,**D**) Representative immunoblots. (**B**,**C**,**E**) Densitometric analysis of fibronectin, collagen I, and CTGF normalized to β-actin. * *p <* 0.05, significantly different from the control group. Data are expressed as mean ± SD (n = 4). ^#^
*p <* 0.05, significantly different from the cisplatin-only group. (**F**) mRNA levels of CTGF examined by quantitative RT-PCR. Data were normalized to GAPDH and expressed as fold changes (n = 3).

**Figure 4 cells-11-00712-f004:**
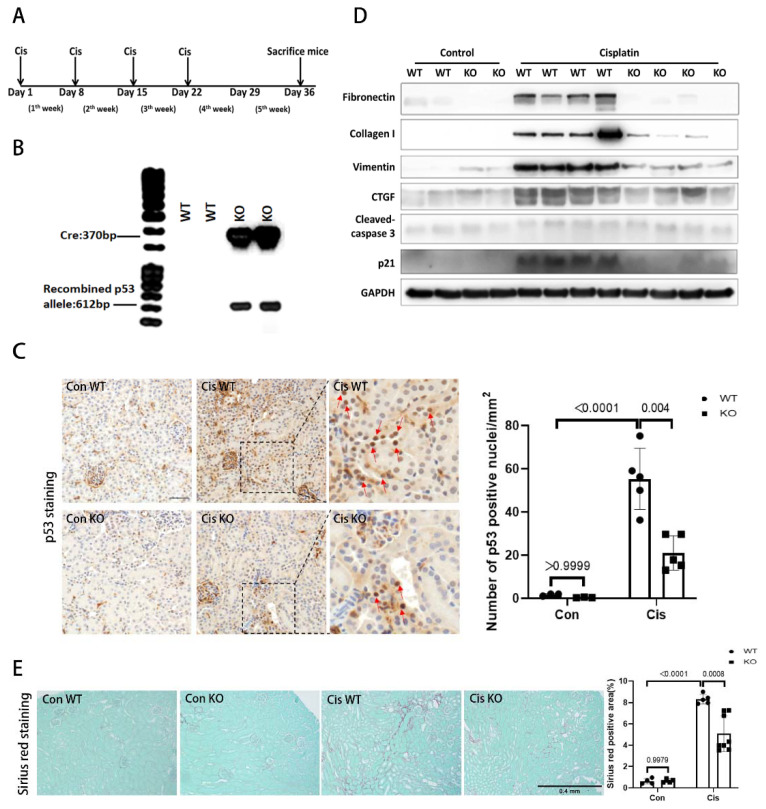
RLDC-induced renal fibrosis is attenuated in PT-p53-KO mice. PT-p53-KO mice and wild-type (WT) littermates were given 4 weekly injections of 8 mg/kg cisplatin to collect samples on day 36 after the first cisplatin injection (**A**). (**B**) Representative DNA gel images verifying p53 gene recombination or deletion in PT-p53-KO kidneys. Genomic DNA from kidney cortex was amplified by PCR to detect the CRE recombinase and recombined p53 allele using specific primers. (**C**) Representative image and quantification of p53 staining in kidney cortical tissues of PT-p53-WT and KO mice. Scale bar = 50 µm. (**D**) Representative immunoblots. Tissue lysates were extracted from the kidney cortex in the PT-p53-KO and PT-p53-WT mice for immunoblot analysis of fibronectin, collagen I, vimentin, CTGF, cleaved-caspase3, p21, and GAPDH. (**E**) Analysis of fibrosis by Sirius Red staining with quantitative analysis of positive staining areas. Scale bar = 0.4 mm. Cis, cisplatin. Con, control. WT, PT-p53-WT. KO, PT-p53-KO.

**Figure 5 cells-11-00712-f005:**
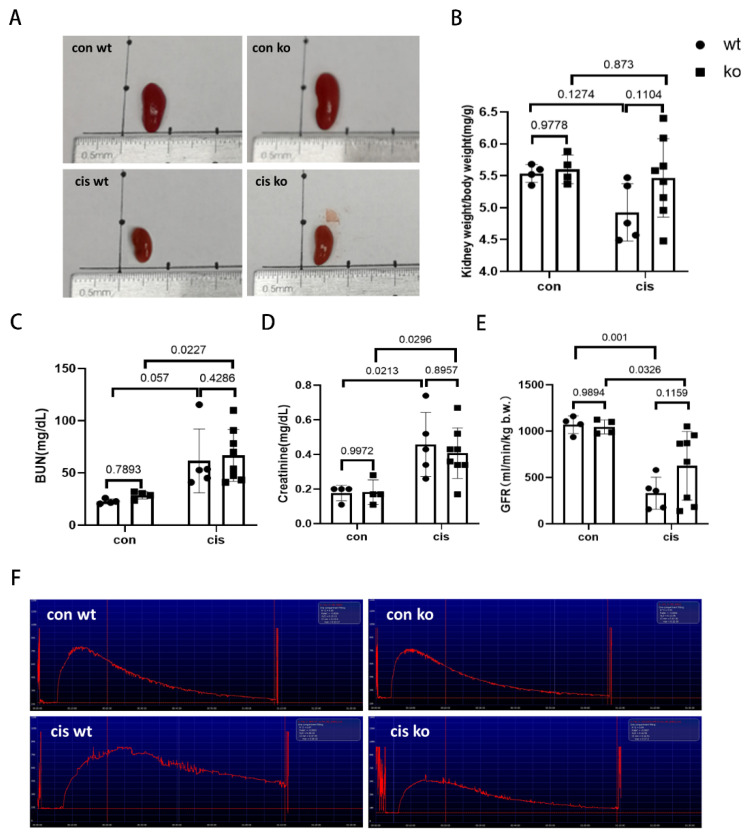
RLDC-induced kidney atrophy and renal function decline are moderately suppressed in PT-p53-KO mice. PT-P53-WT and KO male mice were given 4 weekly injections of 8 mg/kg cisplatin to collect samples on day 36 after the first cisplatin injection. GFR was measured the day before animal sacrifice. Blood samples were collected to monitor blood urea nitrogen (BUN) and serum creatinine levels at the end point. (**A**) Representative images showing kidney size. (**B**) Quantitative analysis of kidney weight/body weight ratio. (**C**) BUN. (**D**) Serum creatinine. (**E**) GFR. (**F**) Representative tracings of GFR measurement by monitoring FITC-sinistrin clearance. Quantitative data are expressed as means ± SD (n = 4–8). Cis, cisplatin. Con, control. WT, PT-p53-WT. KO, PT-p53-KO.

**Figure 6 cells-11-00712-f006:**
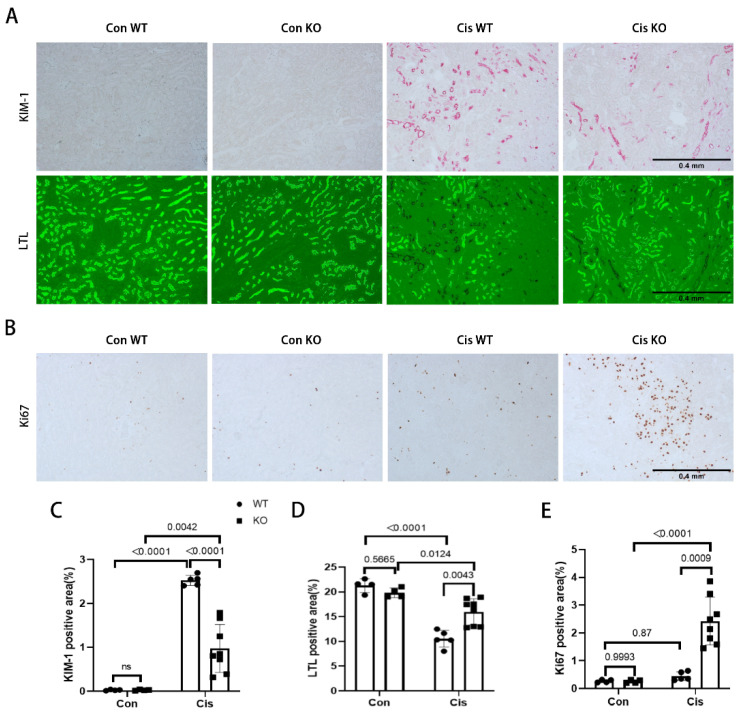
Proximal tubular p53 deficiency suppresses tubular degeneration and induces renal cell proliferation in RLDC-treated mice. (**A**) Representative images of KIM-1 immunohistochemical staining co-stained with LTL. (**B**) Representative images of Ki67 immunohistochemical staining. (**C**–**E**) Quantitative analysis of KIM-1 immunohistochemical staining, LTL staining, and Ki67 immunohistochemical staining. Cis, cisplatin. Con, control. WT, PT-p53-WT. KO, PT-p53-KO.

**Figure 7 cells-11-00712-f007:**
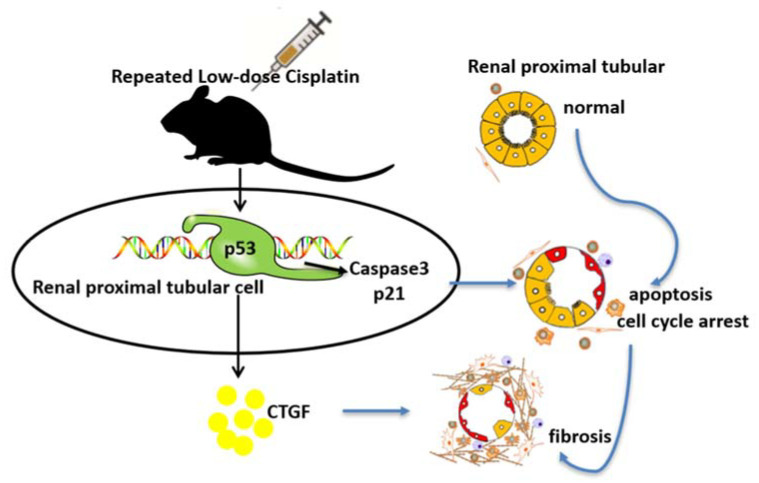
Schematic diagram depicting p53 in chronic kidney problems after repeated low-dose cisplatin (RLDC) treatment. p53 is activated in response to RLDC treatment. Following activation, p53 contributes to tubular cell apoptosis, cell cycle arrest, and the development of a pro-fibrotic phenotype for renal fibrosis.

## Data Availability

All data needed to evaluate the conclusions or reperform analyses in this paper are presented in the manuscript.

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
