# Peer review of "p53 in Proximal Tubules Mediates Chronic Kidney Problems after Cisplatin Treatment"

_cells, 2022, doi:10.3390/cells11040712_

Round 1
Reviewer 1 Report
References should be cited in some sentences between Line 417-424.
Author Response
As suggested, we have added 2 references in the sentences between Line 417-424.
Reviewer 2 Report
Accepted in its present form
Author Response
Thanks for supporting our work.
This manuscript is a resubmission of an earlier submission. The following is a list of the peer review reports and author responses from that submission.
Round 1
Reviewer 1 Report
Thank you very much for giving me the opportunity to review the interesting paper.
2.2: Anti-tumor effect in p53-KO mice should be discussed in “Discussion”. (The anti-tumor effect should be significantly reduced in p53-KO mice compared with WT mice.)
Figure1: It is stated that both of p53 and p21 activation is suppressed in the anti-p53 (pifithrin-α) group.
In the non-cisplatin group,p21 appears to be more activated in the anti-p53 group than in the non-anti-p53 group and the result is opposite to that of the cisplatin group (E). You should explain it in details.
I think the authors' goal is to apply cisplatin clinically as an anti-tumor agent while protecting renal function.
Therefore, administration of anti-p53 may be effective in protecting renal function, but cisplatin may not play a role as an anti-tumor agent. Discussion needs to be provided to addressing these concerns. (For example, the possibility of non-p53-mediated anti-tumor effects of cisplatin etc)
The abbreviation for CTGF should be listed in Line 373 instead of Line 380.
It is recommended to show in the figure how cisplatin, p53, CTGF,p21 are involved in renal function.
We wish you the best of luck in your research.
Reviewer 2 Report
The current work showed the importance of p53 signaling in AKI to CKD model introduced by repeated cisplatin administration using comprehensive in vivo and in vitro experimental models. Though p53 signaling is widely analyzed in various experimental rodent models so far as authors described in introduction and discussion, it is novel that the proximal tubular specific p53-KO mice applied for repeated-cisplatin model.
However, there are several concerns for this work as follows.
Major
- One criticism for this work is lacking a more quantitative approach to elucidate whether p53 expression was truly diminished in this PT-specific p53-KO model. Authors only showed the brief immunofluorescence pictures for p53, but it is not convincing nor quantitative. It is really crucial parts for this work. Authors have to show the precise localization where p53 predominantly express in this mouse model by dual or triple staining of adequate markers like KIM-1 etc, and then have to show the diminishment of p53 in these populations. The analysis of partial (or complete in some cases) reduction of p53 protein expression in WB is also necessary.
- The number of Ki67-positive cells was increased in PT specific p53-KO mice than WT mice with cisplatin, and authors argue that it is attributed by genetic deletion of p53. Generally, more severe injury leads larger number of cells undergoing proliferation at recovery phase for repair, which is opposite results to current work by authors. Ki67 is a marker of cell population entering cell cycle, therefore this discrepancy cannot be explained by p53-mediated cell cycle arrest. (G2/M arrested cells are also positive for Ki67). The authors have to carefully interpret these results including the relationship between p53-deletion and cell proliferation. Additional supportive experiments for detailed interpretations might be required.
- In vitro data showed that p53 inhibition exhibited the antiapoptotic effect. Is the number of apoptotic proximal tubular epithelial cells in p53-KO mice less in vivo at the acute phase after last injection? Cleaved caspase 3 expression is similar between p53-KO and WT with cisplatin, but the timing of sacrifice is chronic phase after injury, and it does not reflect the apoptosis at acute phase, faithfully.
- For mimicking the repeated cisplatin model in vivo, authors performed in vitro experiments of repeated low dose cisplatin treatment for proximal tubular epithelial cell line. Obviously, the dose of cisplatin is lower than previously published investigations in vitro, however, authors have to clarify the difference of cellular responses between single and multiple cisplatin treatment. Is there any difference of gene or protein expression between single high dose vs multiple low dose cisplatin administration? Otherwise, authors have to refer any previous reports using similar protocol.
- Compared with in vivo experiments, the intervals of cisplatin administration in in vitro experiments are shorter, it is also problematic in this work. As mentioned in comment #3 and #4, authors have to clarify whether the results of in vitro experiments truly can apply to the in vivo experiments.
Reviewer 3 Report
The abstract should have quantitative data, check.
The origin source for the utilized instruments and techniques should be mentioned.
Figure 1 (D, E) should be redrawn with high resolution.
Round 2
Reviewer 1 Report
Thank you very much for giving me the opportunity to review your revised paper.
I could not understand the sentence Line 422-423.
Please present results or cite the papers to support the conclusion.
Reviewer 2 Report
For the author
As described in a previous comment for this manuscript, the major criticism for this work is lacking a more quantitative approach to elucidate that the p53 expression was truly diminished in this experimental model. Authors answered that they performed genotyping for p53-KO, however, this data is just a genotype and does not guarantee the precise localization and degree of p53 deletion in this mouse model. In addition, the authors argued that these data have been confirmed in a similar article in JASN 2014, but repeated cisplatin injection model is a different AKI to CKD model, and the evidence for involvement of p53 in this mouse model is still lacking. Therefore, we again strongly recommend to confirm the upregulation of p53 expression and the genetic deletion at earlier phase after last cisplatin injection (e.g. 3days after last injection as described JASN 2014) in this mouse model.